# Effect of prednisolone on glyoxalase 1 in an inbred mouse model of aristolochic acid nephropathy using a proteomics method with fluorogenic derivatization-liquid chromatography-tandem mass spectrometry

Shih-Ming Chen[1¤]*, Chia-En Lin[1¤], Hung-Hsiang Chen[1¤], Yu-Fan Cheng[1¤], Hui-Wen Cheng[1¤], Kazuhiro Imai[2]

1 School of Pharmacy, Taipei Medical University, Taipei, Taiwan, 2 Research Institute of Pharmaceutical Sciences, Musashino University, Tokyo, Japan

¤ Current address: Taipei Medical University, Taipei, Taiwan
* smchen@tmu.edu.tw

## Abstract

Prednisolone is involved in glucose homeostasis and has been used for treatment for aristolochic acid (AA) nephropathy (AAN), but its effect on glycolysis in kidney has not yet been clarified. This study aims to investigate the effect in terms of altered proteins after prednisolone treatment in a mice model of AAN using a proteomics technique. The six-week C3H/He female mice were administrated AA (0.5 mg/kg/day) for 56 days. AA+P group mice were then given prednisolone (2 mg/kg/day) via oral gavage for the next 14 days, and AA group mice were fed water instead. The tubulointerstitial damage was improved after prednisolone treatment comparing to that of AA group. Kidney homogenates were harvested to perform the proteomics analysis with fluorogenic derivatization-liquid chromatography-tandem mass spectrometry method (FD-LC-MS/MS). On the other hand, urinary methylglyoxal and D-lactate levels were determined by high performance liquid chromatography with fluorescence detection. There were 47 altered peaks and 39 corresponding proteins on day 14 among the groups, and the glycolysis-related proteins, especially glyoxalase 1 (GLO1), fructose-bisphosphate aldolase B (aldolase B), and triosephosphate isomerase (TPI), decreased in the AA+P group. Meanwhile, prednisolone decreased the urinary amount of methylglyoxal (AA+P: 2.004 ± 0.301 μg $vs$. AA: 2.741 ± 0.630 μg, $p < 0.05$), which was accompanied with decrease in urinary amount of D-lactate (AA+P: 54.07 ± 5.45 μmol $vs$. AA: 86.09 ± 8.44 μmol, $p < 0.05$). Prednisolone thus alleviated inflammation and interstitial renal fibrosis. The renal protective mechanism might be associated with down-regulation of GLO1 via reducing the contents of methylglyoxal derived from glycolysis. With the aid of proteomics analysis and the determination of methylglyoxal and its metabolite-D-lactate, we have demonstrated for the first time the biochemical efficacy of prednisolone, and urinary methylglyoxal and its metabolite-D-lactate might be potential biomarkers for AAN.

**Data Availability Statement:** All relevant data are within the manuscript and its Supporting Information files.

**Funding:** We are grateful to the financial support from the Cathay General Hospital (108CGH-TMU-06).

**Competing interests:** The authors have declared that no competing interests exist.

## Introduction

Aristolochic acid nephropathy (AAN) was first introduced in 1993 [1]. After the Belgian women ingested slimming pills containing aristolochic acid (AA), their renal function dramatically decreased and managed by dialysis. The feature of AAN is rapid progression into interstitial renal fibrosis and end-stage renal disease [1, 2]. AA is extracted from the *Aristolochiaceae* species and was used for anti-inflammatory activities in traditional medicine. AA is a mixture of 8-methoxy-6-nitrophenanthro-(3,4-D)-1,3-dioxolo-5-carboxylic acid (aristolochic acid I [AAI]) and its 8-demethoxylated form (aristolochic acid II [AAII]) [3]. AAI shows stronger nephrotoxicity than AAII in AAN because of the *O*-methoxy group at position C-8 of the nitrophenanthrene ring [4]. This structure of AAI facilitates AA interactions with DNA, and this AA-DNA adduct leads to cytotoxicity and carcinogenicity. A recent study indicates that AA-induced upper tract urothelial cancer is related to p38 and extracellular signal regulated kinases (ERK) sub-pathways [5]. Despite prohibition of AA-containing herbs, patients are still suffering from AAN [6]. Vanherweghem *et al.* successfully treated AAN with prednisolone, because AAN is thought to be related to the immune response, such as infiltration of immune cells into the renal cortex [7]. Recently, Ma *et al.* showed that low-dose prednisone (0.5 mg/kg) is effective at slowing the progression of AAN via the suppression of monocyte chemoattractant protein-1 (MCP-1) and transforming growth factor-β (TGF-β) activities [8].

Prednisolone, one of the glucocorticoids, acts as endogenous cortisol that is released from the hypothalamic-pituitary-adrenal (HPA) axis, particularly in the case of stress or injury [9]. Prednisolone fights inflammation via the regulation of tumor necrosis factor-α (TNF-α), interleukin-1 (IL-1), and interleukin-6 (IL-6) etc, thus, widely applied to treating various inflammation diseases or states, such as allergies, asthma, dermatitis, rheumatic disorders, systemic lupus erythematosus, and autoimmune disorders[10–12]. Recently, Baudoux *et al.* demonstrated that cluster of differentiation $CD4^+$ and $CD8^+$ T-cells regulate immune responses in AA-induced acute tubular necrosis [13]. However, the mechanism of prednisolone treatment for AAN is still unclear. Moreover, prednisolone plays an important role in glucose homeostasis, but the relationship between prednisolone and glycolysis, including the impact of methylglyoxal, a by-product of glycolysis, has never been studied. The proteomics study with fluorogenic derivatization-liquid chromatography/tandem mass spectrometry (FD-LC-MS/MS) was introduced in 2004 [14, 15]. This FD-LC-MS/MS method was widely applied to screening proteins in cell lines [16], rat [17], mouse models [18]. Therefore, the aim of this study was to explore the effect of prednisolone on changes in glycolysis-related protein expression using FD-LC-MS/MS method in the AAN mice model and clarify the pharmacological mechanisms of prednisolone in AAN model.

## Materials and methods

### Animal experiments

All of the animal use protocols were approved by the Animal Care and Use Committee/Panel of Taipei Medical University (IACUC Approval No: LAC-2013-0282), and the performance was complied with the relevant regulations. The AAN model and experiments protocols were followed previous studies. Six-week-old female C3H/He mice were randomly divided into three groups: normal (N), aristolochic acid (AA), and aristolochic acid + prednisolone (AA+P) groups (n = 10 each group). The AA and AA+P group mice received *ad libitum* access to 3.0 μg/mL of AA-distilled water (0.5 mg/kg/day) orally for 56 days according to the previous study [18]. For the next 14 days, AA+P group were gavage fed prednisolone (2 mg/kg/day),

and AA group were gavage administrated water, respectively. The N group mice drank water during the study period. Mouse urine was collected within 12 hours on days 0, 7, and 14 via a metabolic cage (Tokiwa Chemical Industries Co. Ltd, Japan) [18]. All the mice were sacrificed on day 14. All of the harvested urine, kidney tissue, and blood were stored at -80˚C before analysis [18–20].

## Biochemical assays

All biochemical parameters from each of the 30 mice were measured. Blood urea nitrogen (BUN) was determined using a Beckman blood urea nitrogen kit, and serum creatinine (Scr) was measured with a DxC 600 chemistry analyzer (Beckman Coulter, IN USA) [18]. The fluorometric method was used to determine the activity of urinary NAG, which was defined as the production of 4-MU from 4-MU-NAG in 100 mM citrate buffer (pH 4.6–5.0) within 15 minutes, and the activity was measured at 370 nm/460 nm (excitation/emission wavelength) [21]. Urinary protein was determined via the Bradford method [22]. All the work of this study was shown in Fig 1.

## Histological examination

All of the kidneys from the 30 mice were embedded with paraffin, and the sections were sliced into 4 to 5 μm sections [18, 19]. The kidney sections were stained with periodic acid–Schiff (PAS) (395B, Sigma-Aldrich, Inc., MO, USA) and Masson trichrome stain (HT15, Sigma-Aldrich, Inc., MO, USA) according to the manufacturers' instructions.

## Immunofluorescence study

Samples from kidneys from the 30 mice were prepared for immunofluorescence study which were performed previously [18]. Briefly, the cryostat sections (4 μm) were incubated with rabbit polyclonal transforming growth factor-beta 1 (TGF-β) antibody (21898-1-AP, Proteintech Group, Inc, IL, USA), rabbit polyclonal matrix metallopeptidase 9 (MMP9) (N-Terminal) antibody (10375-2-AP, Proteintech Group, Inc, IL, USA), or rabbit polyclonal anti-human growth factor (HGF) antibody (ab83760, Abcam, OR, USA) for 30 min at room temperature, and all the primary antibodies were diluted with phosphate buffered saline (PBS) at 1:100. The sections were washed with PBS for three times. Under dark environment, the sections were reacted with tetramethyl rhodamine isothiocyanate (TRITC)-labeled anti-rabbit IgG (T6778, Sigma-Aldrich, Inc., MO, USA) and washed with PBS for three times. The images were deconvoluted and processed using a laser confocal microscope, Olympus FV500 (Tokyo, Japan). The positive area (red) was quantified using FluoView Olympus version 4.0 (Tokyo, Japan) [18].

## Proteomics study

**Sample preparation.**   Six kidney samples from each group (n = 10) were selected for the proteomics study. First, about 50 mg of kidney tissue was homogenized with 300 μL of 10 mM 3-[(3-cholamidopropyl) dimethylammonio] propanesulfonic acid$_{(aq)}$ (CHAPS$_{(aq)}$) and measured the amount of protein using BCA Pierce™ BCA Protein Assay Kit [17, 18].

Each of the homogenate was diluted into 4 mg/mL. The 10 μL of homogenate was reacted with 20 μL of 10 mM ethylenediaminetetraacetic acid disodium salt (EDTA·2Na), 20 μL of 50 mM CHAPS$_{(aq)}$, 20 μL of 2.5 mM tris (2-carboxyethyl) phosphine, 25 μL of 8 M guanidine buffer (pH 8.5), and 5 μL of 140 mM 7-chloro-*N*-[2-(dimethylamino)ethyl]-2,1,3-benzoxadiazole-

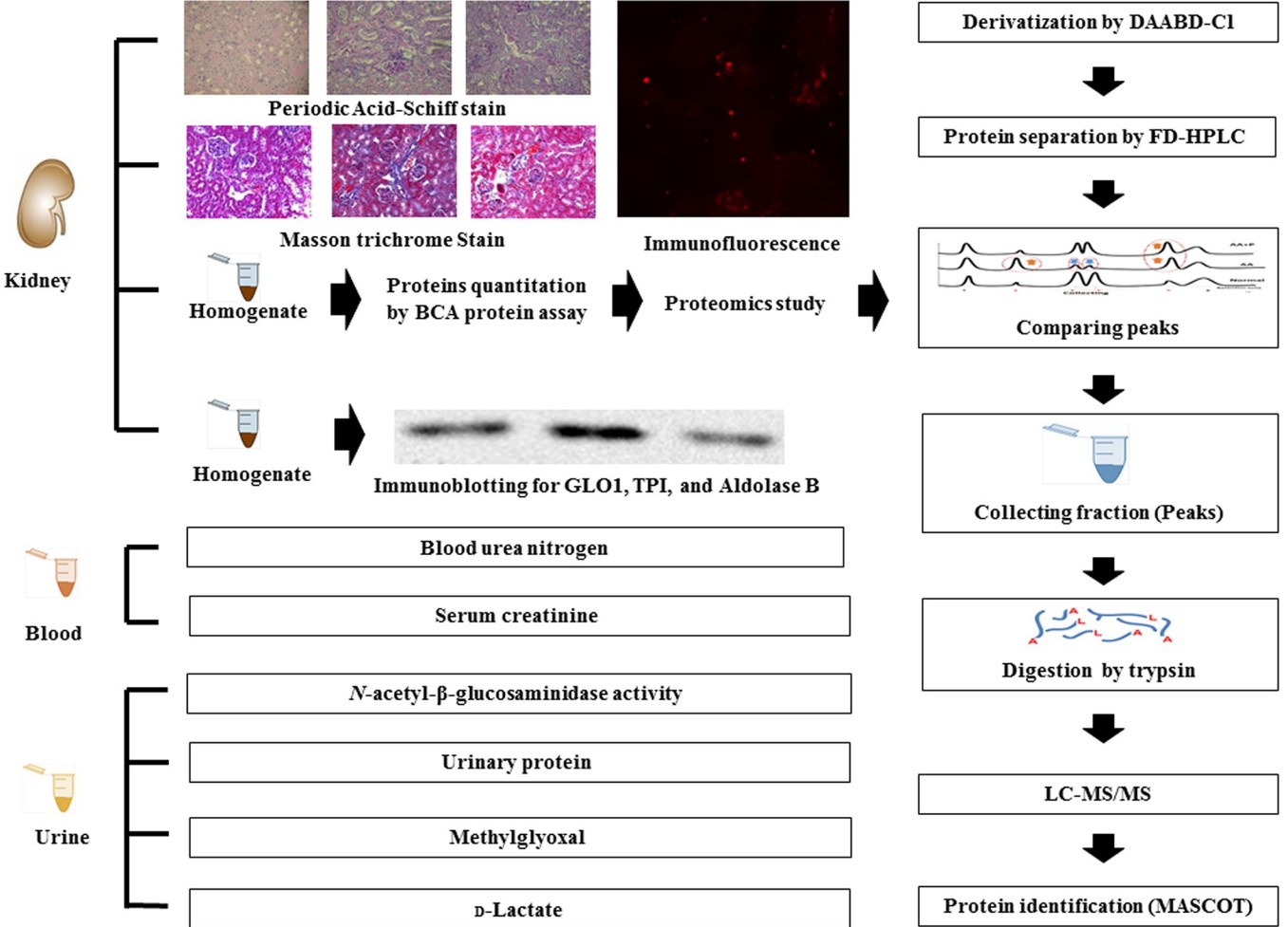

**Fig 1. Flow chart for the current study.** Analysis of kidney tissues: (1) Paraffin-embedded kidney sections were stained with periodic acid–Schiff (PAS) stain and Masson trichrome stain; the cryostat kidney sections underwent immunofluorescence studies in order to determine transforming TGF-β, matrix metallopeptidase 9 (MMP9), and human growth factor (HGF); (2) The kidney samples homogenized with CHAPS (aq) were used for the proteomics study. Before analysis, the proteins were quantitated with the bicinchoninic acid (BCA) protein assay. The kidney sample proteins were derivatized with 4-[2-(dimethylamino)ethylaminosulfonyl]-7-chloro-,1,3-benzoxadiazole] (DAABD-Cl) and then separated by fluorescence detection-high-performance liquid chromatography (FD-HPLC). The altered peak fractions were compared and collected, digested with trypsin, and analyzed by liquid chromatography–tandem mass spectrometry (LC-MS/MS). Finally, the altered proteins were identified with the MASCOT software. (3) Immunoblotting analyses for GLO1, TPI, and aldolase B protein expressions were performed to confirm proteomic study findings. Blood samples were collected to analyze BUN and serum creatinine (Scr). Urine samples were collected to analyze urinary NAG activity and protein, methylglyoxal, and D-lactate levels. Aldolase B, fructose-bisphosphate aldolase B; CHAPs (aq); DAABD-Cl; FD-HPLC, high performance liquid chromatography with fluorescence detection; GLO1, glyoxalase 1; LC-MS/MS, liquid chromatography-tandem mass spectrometry; TPI, triosephosphate isomerase.

4-sulfonamide (DAABD-Cl) in acetonitrile (MeCN). Second, the mixture was derivatized under 40˚C for 10 min and terminated with 3 μL of 20% trifluoroacetic acid (TFA).

**FD-HPLC conditions for protein separation and quantification.** Forty microliters of the reaction mixture (15.5 μg protein) were injected into the FD-HPLC at a flow rate of 0.55 mL/min and separated on with the IMTAKT WX-RP column (250 × 4.6 mm, 3 μm particle size, Imtakt Co., Kyoto, Japan) at 60˚C. Mobile phase A ($H_2O$/MeCN/isopropanol/TFA = 90/9/1/0.15), mobile phase B ($H_2O$/MeCN/isopropanol/TFA = 30/69/1/0.15), and mobile phase C ($H_2O$/MeCN/isopropanol/TFA = 95/4/1/0.20) were used [17, 18]. The gradient program is described in S1 Table. The fraction was monitored at 505 nm (excitation wavelength: 395 nm). Based on the specific retention time of the derivatives, all of the corresponding peaks of the

proteins were quantified by peak height. The FD-HPLC chromatograms of N, AA and AA+P groups were compared by each peak height using Kruskal-Wallis test, and only the altered peak fractions among three groups were manually collected according to the chromatograms for protein identifications.

**Identification of DAABD-derivatized proteins.** Each of the collected peak fractions (altered peaks among three groups) was concentrated to 5 μL under the condition of reduced pressure. In order to identify the DAABD-derivatized proteins by using liquid chromatography tandem-mass spectrometry (LC-MS/MS), the residue from the above step was digested with 20 μL of 50 mM $NH_4HCO_3$ (pH 7.8), 2.5 μL of 10 mM $CaCl_2$, and 2.5 μL trypsin for 2 h at 37˚C. Six microliters of the peptide mixture was directly subjected to LC-MS/MS (Agilent, CA, USA), including NanoLC Agilent 1200, HPLC Agilent 1100, and API 4000Q TRAP. Each sample was loaded onto a nanoprecolumn (Zorbax 300SB-C18; 5 x 0.3mm I.D.; Agilent) in the injection loop and eluted with 0.10% TFA in 2.0% MeCN at 30 mL/min using the Agilent 1100 pump. Then, the peptides were separated in a C18 NanoEase column (75 μm × 100 mm, 3.5 μm particle size; Waters Corporation, CA, USA) at a flow rate of 0.2 μL/min. The mobile phases included mobile phase A ($H_2O$/formic acid [FA] = 99.9/0.1) and mobile phase B (MeCN/FA = 99.9/0.1), and the gradient elusion was performed for 45 min: 1.0% mobile phase B at the beginning; 50% mobile phase B at 31 min; 85% mobile phase B at 33 min; held for 3 min; and returned to 1% mobile phase B at 37 min. All of the peptides were sprayed into the mass spectrometry (MS; API 4000Q TRAP; Agilent) via a distal coated fused-silica needle (75 μm tube i.d., 15 μm tip i.d., PicoTip™ Emitter, New Objective, MA, USA). One-second MS/MS scans were conducted on each precursor ion. The detected ions with m/z between 350 and 1250 were fragmented with capillary energies ranging from 1300–2500 V, and the temperature of the interface heater was set at 150˚C.

These collected peptides were identified using MASCOT according to previously published methods [17, 18]. MASCOT version 2.2 against the National Center for Biotechnology Information was used to analyze the data. The searching parameters of MASCOT were set as the following: taxonomy: mouse; enzyme: trypsin; allowing less than one missed cleavage peptide; peptide charge: 1+, 2+, 3+; variable modification: DAABD-thiol reside of cysteine. The peptide tolerance was set at 1.2 Da, and mass tolerance for the MS and the tandem MS ions were set at 0.6 Da. Under the situation, MASCOT scores which were higher than 45 were counted as valid peptides. If there were many proteins which shared found peptides, the protein that had the highest score was determined.

## Immunoblotting analysis

Six kidney samples from each group (n = 10) were selected to perform immunoblotting analysis.Ten microgram of sample proteins of renal homogenate was loaded into each lane and separated on 12% sodium dodecyl sulfate-polyacrylamide gels using an SDS-PAGE system [18]. The proteins on the gels were transferred onto nitrocellulose membranes. Antibodies against glyoxalase 1 (GLO1; GTX105792, GeneTex, Irvine, CA, USA), β-actin (20536–1-AP, Proteintech, Rosemont,IL, USA), fructose-bisphosphate aldolase B (aldolase B, GTX101363, GeneTex, Irvine, CA, USA), triosephosphate isomerase (TPI, GTX104618, Irvine, CA, USA), and Goat anti-rabbit IgG (H+L), HRP conjugate (SA00001-2, Rosemont,IL, USA) were used at a dilution of 1:1000, 1:2000, 1:3000, 1:3000, and 1:4000, respectively. The signals corresponding to the bands of GLO1 and β-actin were measured by the TOOL Sensitive ECL kit. ImageJ was used to quantify the intensity of the bands. The relative GLO1 levels were defined as the ratio of GLO1 to β-actin intensity. All of the information of antibody was described in S3 Table.

## Determination of urinary amount of methylglyoxal

Urinary methylglyoxal amount from all of the mice were determined. The level of urinary methylglyoxal was determined by FD-HPLC according to a previously published method [20, 23–25]. In short, the methylglyoxal was incubated with 6-diamino-2,4-dihydroxypyrimidine sulfate (DDP) for 30 min at 60°C and stopped with 0.01 M citric acid (pH 6.0). Twenty μL of the derivative samples was injected into FD-HPLC, and the flow rate was 0.7 mL/min. The derivatized methylglyoxal was separated with a mobile phase (0.01 M citric acid buffer (pH 6.0)/MeCN = 97/3) using an ODS column (250 × 4.6 mm, 5 μm particle size; Biosil Chemical Co. Ltd., Taipei, Taiwan) at 33°C. These fractions were measured at an emission of 500 nm with an excitation of 330 nm. The amount of urinary methylglyoxal was defined as level of methylglyoxal × 12 h-urinary volume.

## Determination of urinary amount of D-lactate

**Preparation of urine sample.** Urinary D-lactate amount from all of the mice were determined. The column-switching FD-HPLC system was used to determine the levels of urinary D-lactate [19, 26, 27]. Twenty microliter of all urine samples were mixed with 10 μL of propionic acid (as the internal standard [I.S.]) and 170 μL of MeCN before configuration (700 g, 10 min, 4°C). Then, 100 μL of sample supernatants were derivatized with 100 μL of 8 mM 4-nitro-7-piperazino-2,1,3-benzoxadiazole (NBD-PZ) in MeCN, 25 μL of 280 mM 2,2'-dipyridyl disulfide (DPDS) in MeCN, and 25 μL of 280 mM triphenylphosphine (TPP) in MeCN at 30°C for 3 h. Finally, the reaction was stopped with 250 μL of 0.1% $TFA_{(aq)}$, and the derivatives were purified via passing through the MonoSpin™ SCX cartridge (GL Science Inc., Tokyo, Japan).

**Separation of lactate.** The urinary lactate derivatives were separated with the mobile phase ($H_2O$/MeCN/methanol/ = 68/12/20) using an Aqu-ODS-W-5u column (250 × 4.6 mm, 5 μm particle size; Biosil Chemical Co. Ltd, Taipei, Taiwan) at 30°C. The flow rate was set 0.7 mL/min for 0–35 min and 0.9 mL/min for 35.1–60 min [19, 27]. The amount of urinary lactate was defined as lactate levels ×12 h urinary volume.

**Enantiomeric separation of D-lactate.** The fraction of lactate derivatives was collected and introduced into a Chiralpak AD-RH column (150 × 4.6 mm, 5 μm particle size; Daicel Co. Osaka, Japan) with the mobile phase ($H_2O$/MeCN = 40/60) at a flow rate of 0.3 mL/min. Both the total lactate and D-lactate levels were determined according to the areas of the corresponding peaks on the chromatograms (D-7500 integrator; Hitachi, Tokyo, Japan). The derivatives were detected at an emission wavelength of 547 nm with an excitation of 491 nm [19, 27]. Urinary D-lactate amount were defined as levels of D-lactate ×12 h urinary volume.

## Statistical analysis

The results are expressed as means ± standard deviation. The significant difference in means was determined using the Kruskal-Wallis test for nonparametric statistics; $p$-values less than 0.05 was taken to indicate statistical significance. All of the data analysis was performed using Statistical Product and Service Solutions (SPSS) for Windows 19[th] version (IBM, IL, USA).

## Results

### Biochemical assays

There were no significant differences at baseline in NAG (N [1.93 ± 0.15 U/L] *vs.* AA [1.92 ± 0.12 U/L] *vs.* AA+P [1.95 ± 0.10 U/L]) and urinary protein (N [1.21 ± 0.34 mg/day] *vs.* AA [1.17 ± 0.29 mg/day] *vs.* AA+P [1.26 ± 0.34 mg/day]) among the three groups. BUN (20.67 ± 0.73 *vs.* 23.00 ± 2.09 mg/dL, $p < 0.05$), Scr (0.27 ± 0.04 *vs.* 0.36 ± 0.06 mg/dL,

**Table 1. Biochemical parameters of normal (N), aristocholochic acid (AA), and aristocholochic acid + prednisolone (AA+P) groups.**

| Group | NAG (μM/min/L) | | UP (mg/day) | | BUN (mg/dL) | Scr (mg/dL) |
|---|---|---|---|---|---|---|
| | Baseline | Day 14 | Baseline | Day 14 | Day 14 | Day 14 |
| N | 1.93 ± 0.15 | 1.93 ± 0.06** | 1.21 ± 0.34 | 1.26 ± 0.20** | 17.85 ± 1.91** | 0.22 ± 0.07** |
| AA | 1.92 ± 0.12 | 2.94 ± 0.13 | 1.17 ± 0.29 | 2.94 ± 0.09 | 23.00 ± 2.09 | 0.36 ± 0.06 |
| AA+P | 1.95 ± 0.10 | 2.22 ± 0.42* | 1.26 ± 0.34 | 1.75 ± 0.68* | 20.67 ± 0.73* | 0.27 ± 0.04* |

There were no significant difference of NAG and UP among N, AA, and AA+P groups at baseline. The BUN, Scr, NAG activity, and UP excretion of N and AA+P group mice significantly decreased on day 14 compared with those of AA-group mice. Baseline was defined as the day at the beginning of experiment (before the mice were administrated AA). N, normal group; AA, aristolochic acid group; AA+P: aristolochic acid + prednisolone; NAG, $N$-acetyl-β-D-glucosamine; BUN, blood urea nitrogen; Scr, serum creatinine; UP, urinary protein.

* $p < 0.05$

** $p < 0.01$ significantly different from the AA group.

$p < 0.05$), NAG (2.22 ± 0.42 *vs.* 2.94 ± 0.13 U/L, $p < 0.05$), and urinary protein (1.75 ± 0.68 *vs.* 2.94 ± 0.09 mg/day, $p < 0.05$) in the AA+P group were significant lower in the AA group on day 14 (Table 1).

## Histological examination

Sections stained with PAS in AA-group exhibited pathological damage, including cell infiltration, tubular cell atrophy, and interstitial fibrosis, but those in AA+P group exhibited alleviation. THIS of AA-group (7.34 ± 0.89) was significantly higher than those of N (0.36 ± 0.16), and AA+P-group (2.99 ± 0.89) on day 14 (Fig 2). The histological examination for the Trichrome indicated collagen deposition in the AA group (13.19 ± 1.96%) was significantly higher than that of the mice in the N (6.31 ± 0.26%) and AA+P groups (9.05 ± 1.24%) ($p < 0.05$) (Fig 3).

## Immunofluorescence study

The semi-quantification scores of TGF-β (N [0.35 ± 0.37] *vs.* AA [6.88 ± 1.18] *vs.* AA+P [2.67 ± 0.57]) were significantly decreased ($p < 0.05$); those of MMP-9 (N [0.63 ± 0.42] *vs.* AA [3.43 ± 1.36] *vs.* AA+P [11.20 ± 0.84]) and HGF (N [0.23 ± 0.51] *vs.* AA [2.81 ± 0.83] *vs.* AA+P [5.79 ± 0.79]) increased in the N and AA+P groups on day 14 ($p < 0.05$) (Fig 4).

## Separation and identification of altered proteins

There were 47 altered peaks among the N, AA, and AA+P groups on day 14 (Fig 5), which were identified according to the MASCOT analysis included 39 proteins, including glycolysis, anti-oxidation, ATP synthesis, and apoptosis-related proteins, etc. The amplified chromatograms of Fig 5 (S1 Fig) and all the altered proteins (S2 Table) can be found in supporting information. All of the chromatograms of each group were shown in S2 Fig. Most of the glycolysis-related proteins in the AA+P group were lower than those in the AA group. These glycolysis-related proteins included fructose-bisphosphate aldolase B (aldolase B), glyoxalase 1 (GLO1), pyruvate dehydrogenase E1 α 1 (PDH E1 α 1), M2-type pyruvate kinase (PKM2), triosephosphate isomerase (TPI), phosphoglycerate kinase (PGK) and aldose reductase (AR) (Table 2). The expression of altered protein was shown in Fig 6.

## Immunoblotting analysis

The differences in GLO1, aldolase B, and TPI expressions were similar among the three groups (Fig 7A, 7B and 7C). On day 14, the GLO1 protein expression in the kidney homogenate of the N

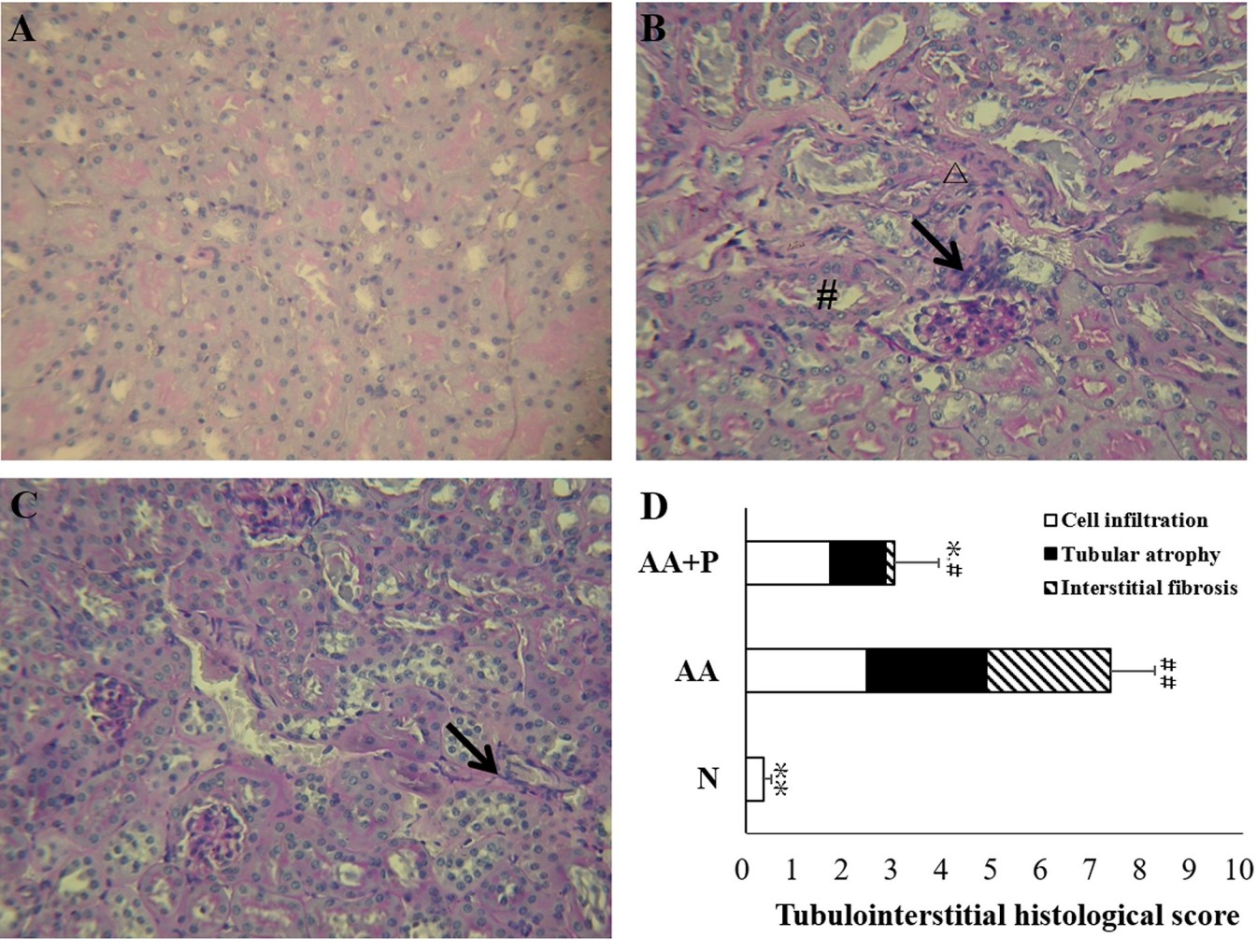

**Fig 2. Periodic acid-Schiff (PAS) staining and tubulointerstitial histological score (TIHS) of kidney on day 14.** There was generally no damage in the renal cortex of normal-group (N) mice (A). Moderate tubulointerstitial damage existed in the renal cortex of AA-group mice (B). AA+P group mice demonstrated amelioration of tubulointerstitial damage (C). (Periodic acid-Schiff [PAS] stain, 200× magnification). The tubulointerstitial histological scores (TIHS, including cell infiltration, tubular atrophy, and interstitial fibrosis) of mice in the AA group were significantly higher than those of mice in the normal and AA+P groups on day 14 (D). N, normal group; AA, aristolochic acid group; AA+P: AA+P: aristolochic acid + prednisolone.Δ indicates cell infiltration; → indicates interstitial fibrosis; # indicates tubular atrophy. $^*p < 0.05$, $^{**}p < 0.01$ significantly different from the AA group; # $p < 0.05$, ## $p < 0.01$ significantly different from the N group.

(100.0 ± 30.9%) and AA+P (108.9 ± 10.7%) groups were significantly lower than those of the AA group (191.60 ± 61.1%) ($p < 0.05$) (Fig 7D), which were similar to the FD-LC-MS/MS proteomics findings. Moreover, aldolase B protein expression in the kidney homogenate of the N (100 ± 36.3%) and AA+P (136.2 ± 54.7%) groups were significantly lower than those of the AA group (223.6 ± 52.5%) ($p < 0.05$) as shown in Fig 7E.TPI protein expression in the kidney homogenate of the N (100.0 ± 19.6%) and AA+P (121.4 ± 9.3%) groups were significantly lower than those of the AA group (181.5 ±25.9%) ($p < 0.05$) (Fig 7F). All the immunoblotting were shown in S3 Fig.

## Amount of methylglyoxal in urine

The amount of urinary methylglyoxal in the AA (3.413 ± 0.596 μg) and AA+P (3.362 ± 1.049 μg) groups were significantly higher than those in the N group (1.561 ± 0.752 μg) on day 0, but those in both N (1.878 ± 0.396 μg) and AA+P (2.004 ± 0.301 μg) groups were significantly lower than those in the AA group (2.741 ± 0.630 μg) on day 14 ($p < 0.05$) (Fig 8).

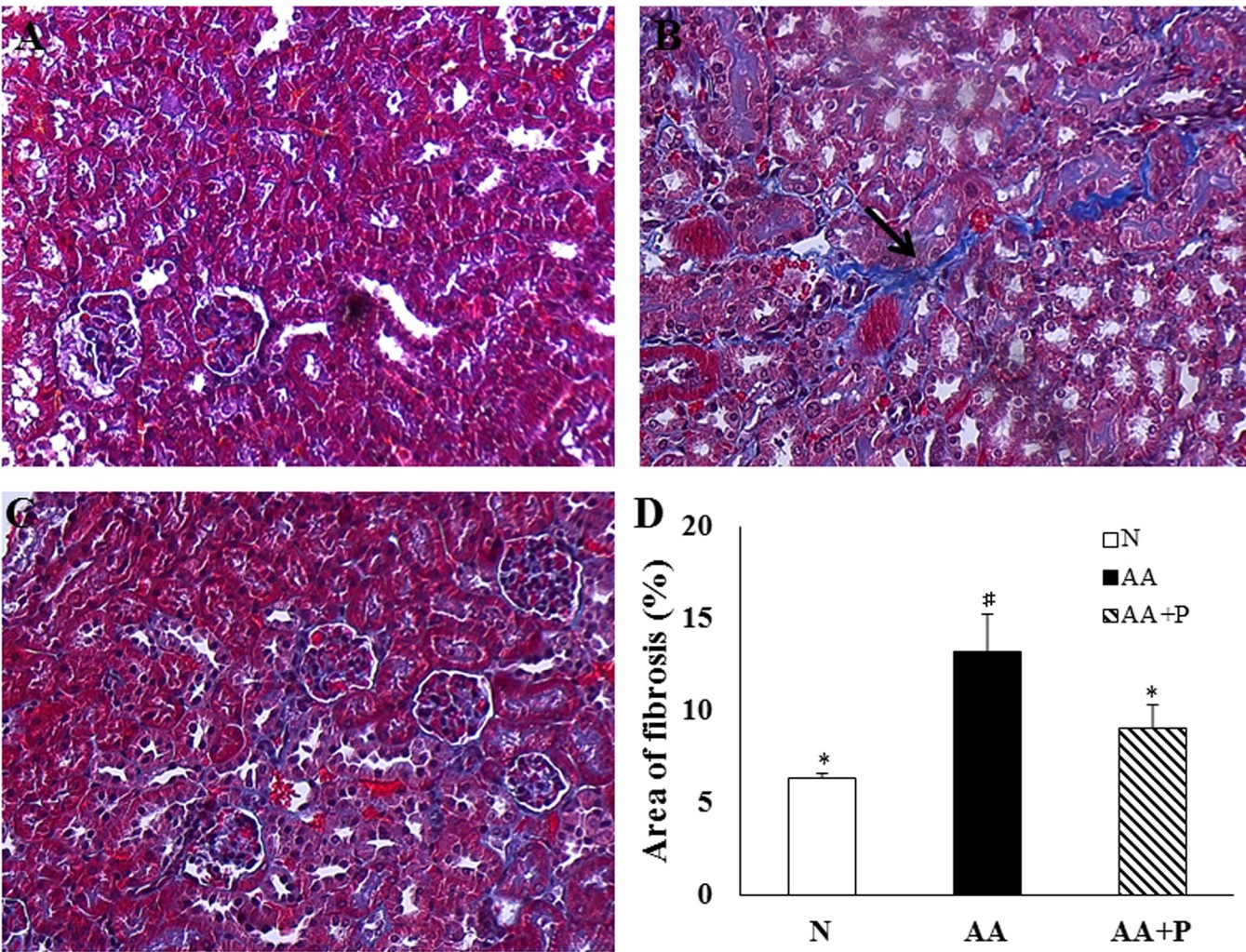

**Fig 3. Masson's trichrome staining and semi-quantification of collagen deposition on day 14.** Collagen deposition in AA-group mice (B) was more severe than that in AA+P-group mice (C). There was almost no collagen deposition in N-group mice (A). The area of fibrosis was assessed based on the blue coloration in the AA group was higher than that in the N and AA+P groups (D). N, normal group; AA, aristolochic acid group; AA+P: AA+P: aristolochic acid + prednisolone. → indicates interstitial fibrosis. * $p < 0.05$ significantly different from the AA group; # $p < 0.05$ significantly different from the N group.

## Amount of total and D-lactate in urine

The amount of urinary total lactate in AA ($1.032 \pm 0.245$ and $0.873 \pm 0.322$ mmol) and AA+P ($1.008 \pm 0.307$ and $0.756 \pm 0.278$ mmol) were significantly higher than that in the N group ($0.250 \pm 0.086$ and $0.429 \pm 0.147$ mmol) on day 0 and 14 ($p < 0.05$) (Fig 8). Moreover, the amount of urinary D-lactate in AA ($105.7 \pm 47.4$ μmol) and AA+P ($129.9 \pm 31.2$ μmol) were significantly higher than that in the N ($3.557 \pm 2.370$ μmol) group on day 0 ($p < 0.05$); but the amount of urinary D-lactate in the N ($10.10 \pm 5.84$ μmol) and AA+P ($54.07 \pm 5.45$ μmol) groups were significantly lower than those in the AA group ($86.09 \pm 8.44$ μmol) on day 14 (Fig 9).

## Discussion

Tubulointerstitial injury, such as interstitial renal fibrosis, tubular cell atrophy, and cell infiltration, was truly induced via administration of AA-distilled water for 56 days as noted in the previous findings [18]. After chronic inflammation, macrophage accumulation leads to fibrosis

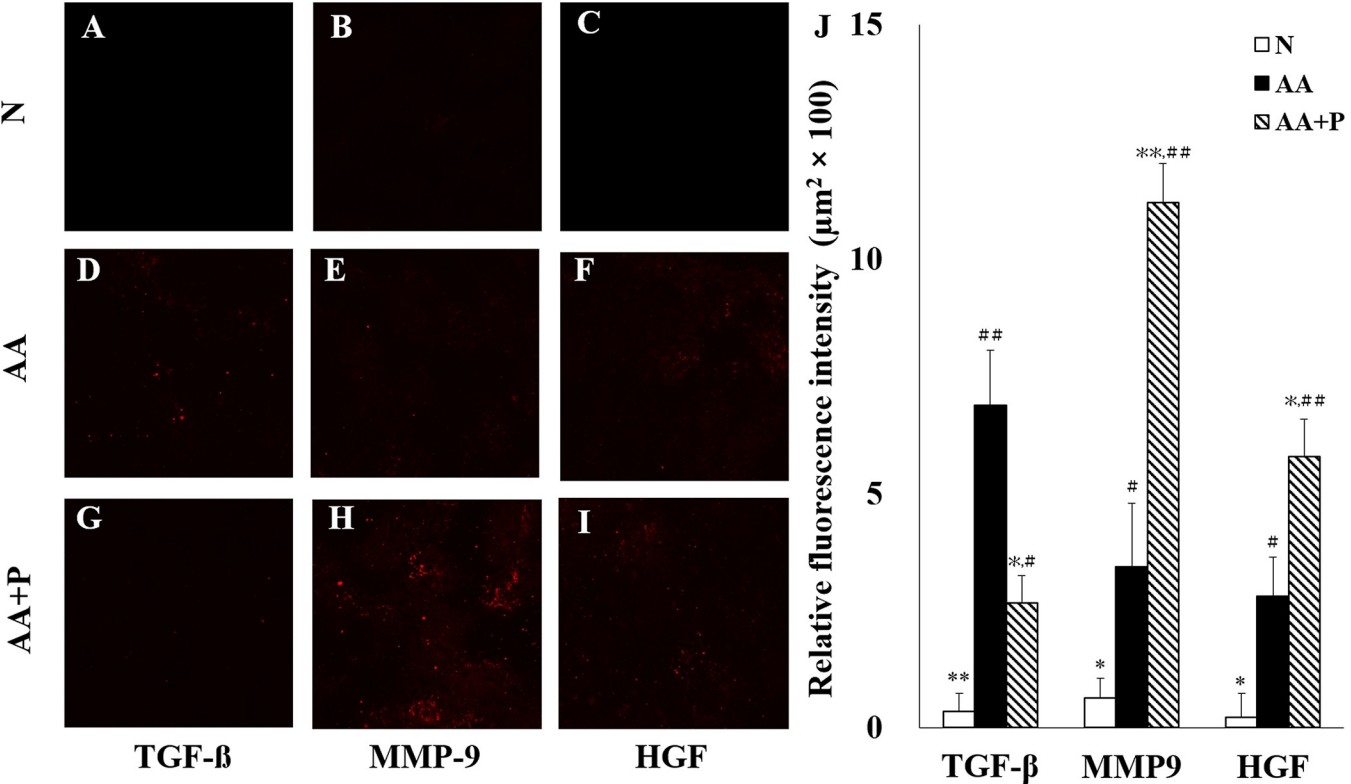

**Fig 4. Immunofluorescence staining and semi-quantification in the tubular interstitium on day 14.** Confocal images (A–I) reveal a red color that demonstrates the deposition of TGF-β (A, D, G), MMP-9 (B, E, H), and HGF (C, F, I) in the N, AA, and AA+P groups, respectively. The expression of TGF-β showed amelioration in AA+P-group mice (G) compared with mice in the AA group (D). The fluorescence intensity of MMP-9 (H) and HGF (I) were significantly increased in AA+P-group mice compared with those in the AA group (E, F). The relative fluorescence intensity of TGF-β, MMP-9, and HGF was semi-quantified (J). The expressions of TGF-β (A), MMP-9 (B), and HGF (C) in the N group were the lowest among the three groups. N, normal group; AA, aristolochic acid group; AA+P: aristolochic acid + prednisolone. TGF-β, transforming growth factor-β; MMP-9, matrix metallopeptidase 9; HGF, hepatocyte growth factor. * $p < 0.05$, ** $p < 0.01$ significantly different from the AA group; # $p < 0.05$, ## $p < 0.01$ significantly different from the N group.

via activation of the fibrosis-related myofibroblasts which release extracellular matrix to repair the damage tissue. These histological results differed from the findings in other acute model which was acute tubular necrosis caused by short-term administration of high-dose AA (5–10 mg/kg/day) [19, 20]. However, the long-term use of low-dose AA induces interstitial renal fibrosis, which is similar to human AAN as noted in the first AAN report [1].

Consistent with the finding by Vanherweghem *et al.*, this study also demonstrated the efficacy of prednisolone for AAN, which supported by improvement of tubular damage and collagen deposition, as well as immune markers (decreased in TGF-β expression and increased HGF and MMP-9 expression). In order to explore prednisolone-induced proteins, this study was the first research to use the proteomics study with FD-LC-MS/MS method to screen proteins after prednisolone treatment, demonstrating that most of the glycolysis-related proteins increased in the AA group and decreased in the AA+P group. Expression of ATP-related proteins also showed similar to glycolysis-related proteins. This might be related to ATP consumption and glycolysis activation due to renal damage and mitochondrial permeability transition pore [20, 28, 29]. After renal damage, dysfunctions of mitochondrial homeostasis and ATP consumption lead to ATP depletion in acute ischemic kidney injury and diabetic nephropathy [30, 31]. Glycolysis, which harvests energy for repair and regeneration of proximal tubule epithelial cells, are composed of three steps. First, glucose converts into fructose

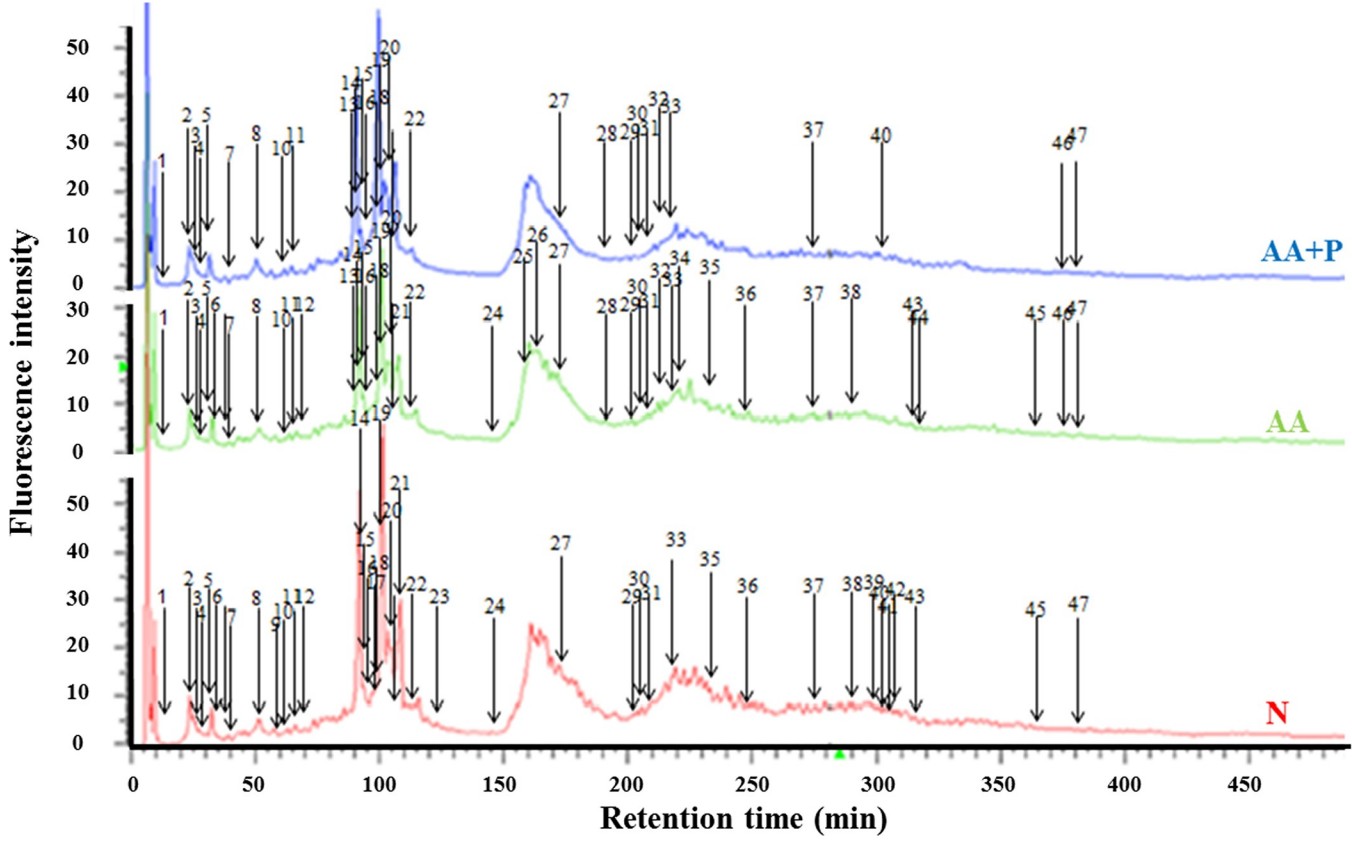

**Fig 5. Chromatograms of proteins in the kidneys of mice derivatized with DAABD-Cl on day 14.** The lower, middle, and upper chromatograms were obtained from the kidney homogenates of normal- (red), AA- (green), and AA+P- (blue) group mice, respectively. The 47 altered peaks among three groups were numbered. N, normal group; AA, aristolochic acid group; AA+P: aristolochic acid + prednisolone; DAABD-Cl, 7-chloro-*N*-[2-(dimethylamino)ethyl]-2,1,3-benzoxadiazole- 4-sulfonamide.

**Table 2. Comparison of glycolysis-related proteins among the N, AA, and AA+P groups on day 14.**

| Peak number[a] | N: AA: AA+P (Ratio) | Protein | Molecular mass (Da) | GI number |
|---|---|---|---|---|
| 22 | 1: 6.56[#]: 0.87[*] | Fructose-bisphosphate aldolase B | 39,548 | gi\|15723268 |
| 22 | 1: 6.56[#]: 0.87[*] | Glyoxalase 1 | 20,826 | gi\|19354350 |
| 22 | 1: 6.56[#]: 0.87[*] | Pyruvate dehydrogenase E1 α 1 | 43,204 | gi\|6679261 |
| 28 | 1: 13.40[#]: 4.06 | M2-type pyruvate kinase | 57,878 | gi\|1405933 |
| 32 | 1: 39.00[#]: 1.85[*] | Triosephosphate isomerase | 26,679 | gi\|54855 |
| 32 | 1: 39.00[#]: 1.85[*] | Phosphoglycerate kinase | 59,716 | gi\|6679937 |
| 32 | 1: 39.00[#]: 1.85[*] | Aldose reductase | 35,725 | gi\|786001 |

[a]The peak numbers correspond to those shown in Fig 4.

[b]The ratio of AA or AA+P groups to N group is listed in the table; the intensity of the N group was set at 1.

[c]NCBI processed each consecutive sequence record as GI number, a simple series of digits. N, normal group; AA, aristolochic acid group; AA+P: aristolochic acid + prednisolone.

[*]$p < 0.05$, significantly different from the AA group

# $p < 0.05$, significantly different from the N group.

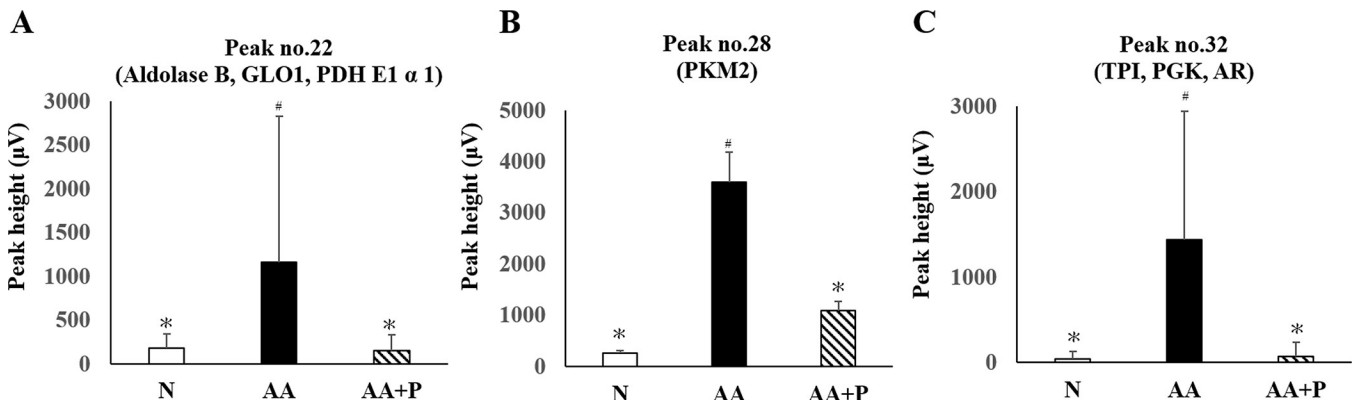

**Fig 6. Altered peak heights corresponding to the glycolysis-related proteins.** The altered peak no. 22, 28 and 32 were compared in Fig 4 and identified glycolysis-related proteins by MASCOT. Peak numbers are the same as in Table 1. Aldolase B: fructose-bisphosphate aldolase B; GLO1: glyoxalase 1; PDH E1 α 1: pyruvate dehydrogenase E1 α 1; PKM2: M2-type pyruvate kinase; TPI: triosephosphate isomerase; PGK: phosphoglycerate kinase; AR: aldose reductase. * $p < 0.05$ significantly different from the AA group; # $p < 0.05$ significantly different from the N group.

1,6-bisphosphate via phosphorylation. Moreover, aldose reductase converts some of glucoses into sorbitol which is one of the source of fructose 1,6-bisphosphate [32]. Second, fructose 1,6-bisphosphate is cleaved into glyceraldehyde 3-phosphate (GAP) and dihydroxyacetone phosphate (DHAP) catalyzing by aldolase B [33]. TPI catalyzes the isomerization of DHAP to GAP which is on the direct pathway of glycolysis, and some of DHAP metabolize into methyl-glyoxal by methylglyoxal synthase [34, 35]. Finally, PGK, pyruvate kinase, and other enzymes successively involve in oxidation of GAP until production of pyruvate and ATP [35, 36]. Moreover, pyruvate dehydrogenase catalyzes pyruvate to acetyl-CoA which enter into Kreb cycle for energy production [37].

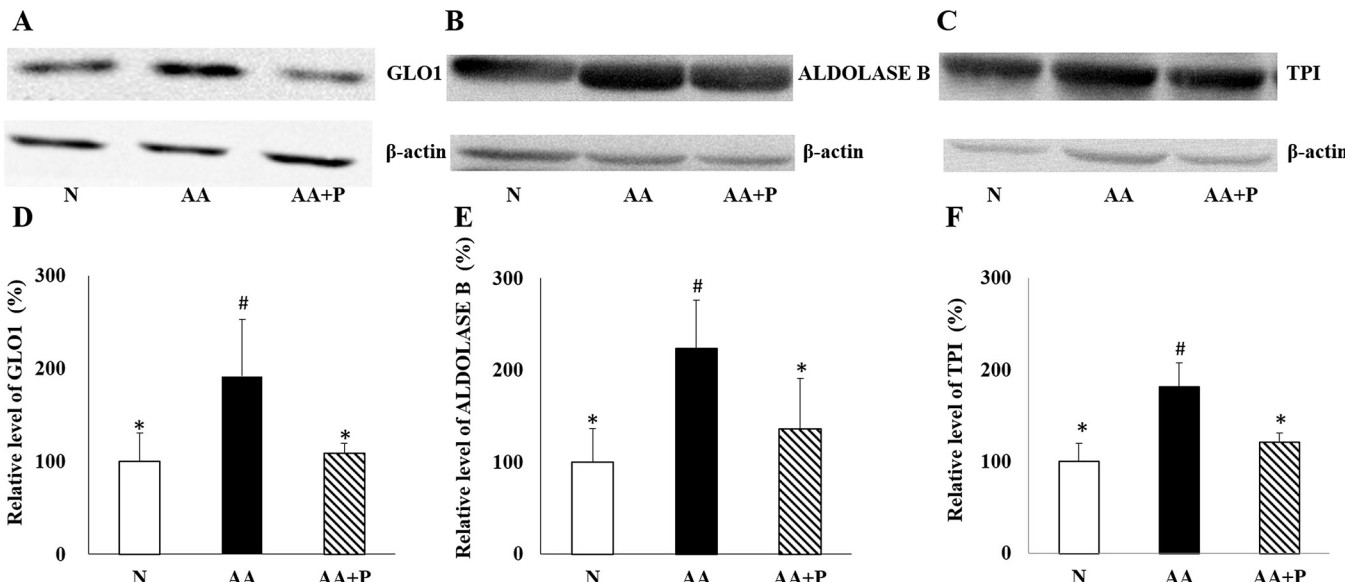

**Fig 7. Immunoblotting analysis of GLO1, aldolase B, and TPI in kidney homogenates on day 14.** Expression of GLO1 (A), aldolase B (B), and TPI (C) among the three groups on day 14. The β-actin was used as the internal standard. Semi-quantitation of the relative GLO1 (D), aldolase B (E), and TPI (F) level. The expression of relative GLO1, aldolase B, and TPI level decreased in the AA+P group compared with the AA group. N, normal group; AA, aristolochic acid group; AA+P: aristolochic acid + prednisolone; GLO1, glyoxalase 1; aldolase B,fructose-bisphosphate aldolase B;TPI, triosephosphate isomerase. * $p < 0.05$ significantly different from the AA group; # $p < 0.05$ significantly different from the N group.

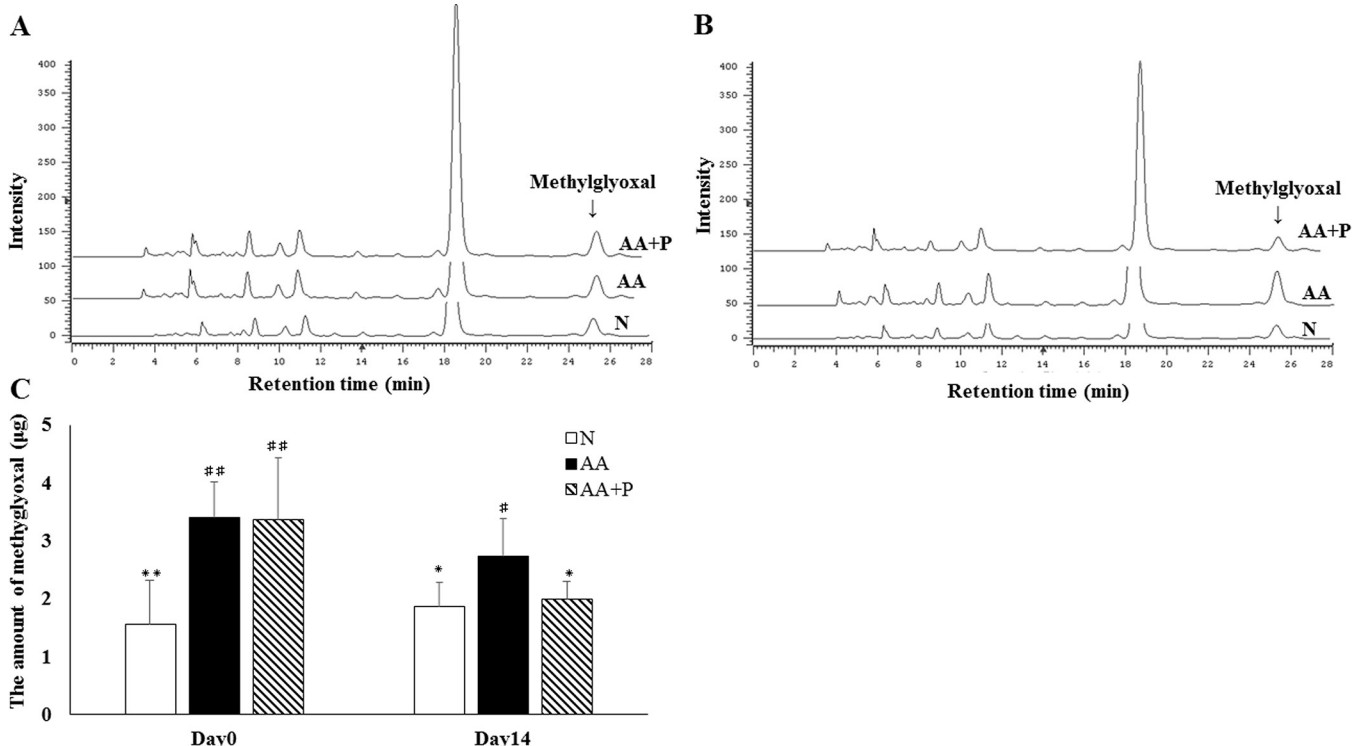

**Fig 8. The amount of methylglyoxal in the urine on days 0 and 14.** The amount of urinary methylglyoxal in the N group was lower than that in the AA group on days 0 and 14. There was no significance in the amount of urinary methylglyoxal between the AA and AA+P groups on day 0, but the amount of urinary methylglyoxal in the AA+P group was significantly lower than that in the AA group on day 14. N, normal group; AA, aristolochic acid group; AA+P: aristolochic acid + prednisolone. $*$ $p < 0.05$, $**$ $p < 0.01$ significantly different from the AA group; # $p < 0.05$, ## $p < 0.01$ significantly different from the N group.

Several studies have proven the relationship between prednisolone and glucose homeostasis, particularly gluconeogenesis and insulin resistance, but the essential glycolysis-related protein, GLO1, was found in the present study. GLO1 is the enzyme of glyoxalase system which is the major pathway to detoxify methylglyoxal [38]. GLO1, the rate-limiting enzyme, spontaneously catalyses the conversion of methylglyoxal-GSH hemithioacetal into thioester $S$-D-lactoylglutathione [39]. This step is the most important in metabolism of methylglyoxal into D-lactate [39, 40]. GLO2 metabolizes $S$-D-lactoylglutathione to D-lactate and GSH. The renal protective effect of GLO1 has been proven in different models of kidney injury, and the down-regulation of GLO1 expression exacerbates the renal function [41–43]. Kumagai *et al.* indicated that the GLO1 protein expression in sham group is similar to those in ischemia/reperfusion injury group in an acute rat model [41], but the change of D-lactate has not been determined. However, methylglyoxal, D-lactate, and GLO1 simultaneously increased after 56 days of AA administration in this study. The different findings suggest that the duration of the study period and the severity of tubular injury might impact on GLO1 expression. First, although the mechanism of GLO1 induction is unclear, enzyme induction is a time-consuming process. Second, GLO1, which exists in the cytosol of each cell, might be depleted if the death of tubular epithelial cells occurs during the progression of necrosis.

Methylglyoxal, one of by-products from glycolysis, leads to cell apoptosis and cytotoxicity due to reactive carbonyl groups which reacts with proteins and nucleic acids. These methylglyoxal-adducts are called advanced glycation end products (AGEs) and lose their function [44–46]. Several studies have indicated that methylglyoxal and D-lactate increase under

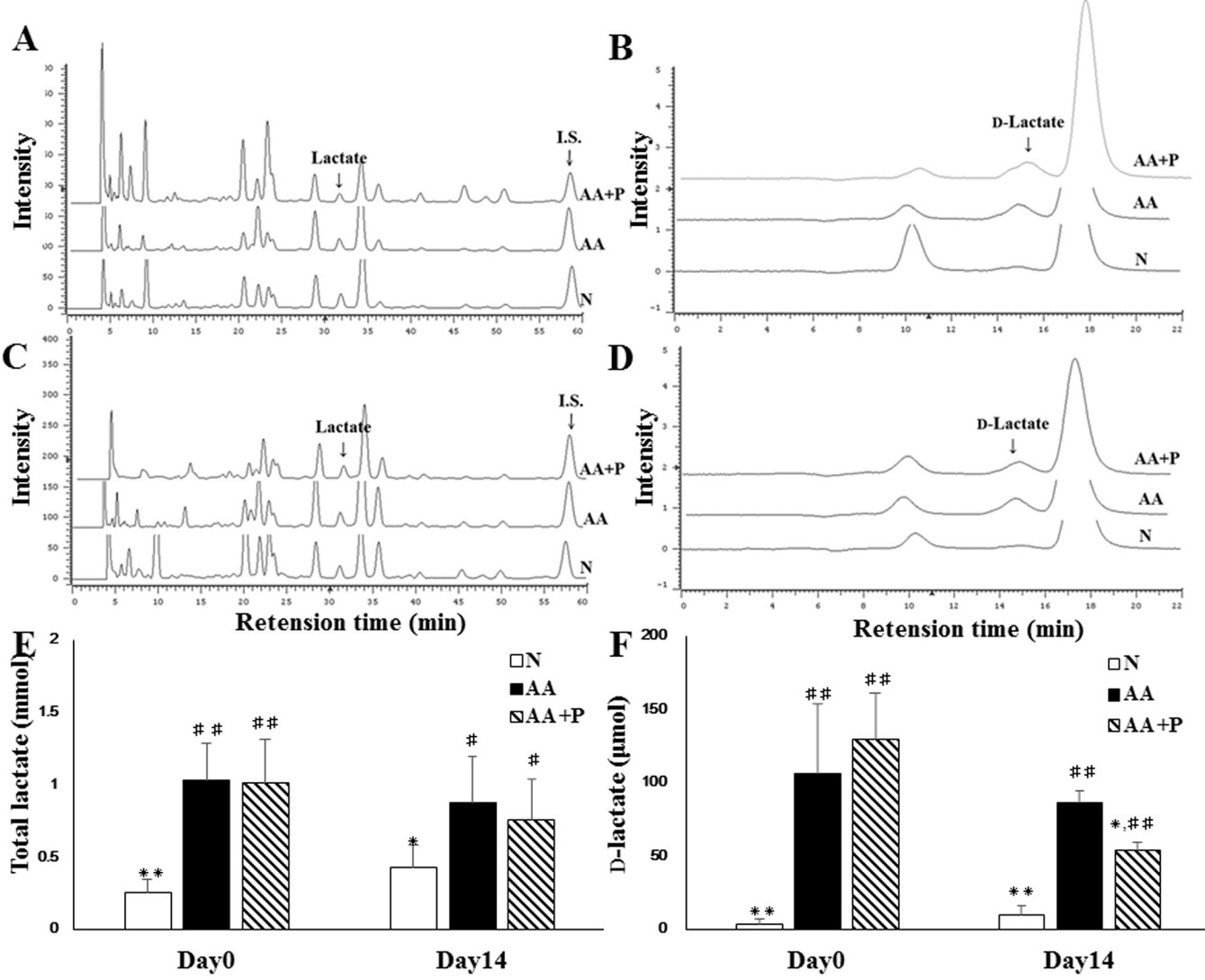

**Fig 9. The amount of total lactate and D-lactate in the urine on days 0 and 14.** The chromatograph of total lactate separation on days 0 (A) and 14 (C). The chromatograph of D-lactate separation on days 0 (B) and 14 (D). The amount of urinary total lactate in the N group was lower than that in the AA and AA+P groups, and there was no significance between the AA and AA+P groups on days 0 or 14 (E). The amount of urinary D-lactate in the N group was lower than that in the AA and AA+P groups on days 0 and 14. There was no significance in the amount of urinary D-lactate between the AA and AA+P groups on day 0, but the amount of urinary D-lactate in the AA+P group was significantly lower than that in the AA group on day 14 (g). N, normal group; AA, aristolochic acid group; AA+P: aristolochic acid + prednisolone. $^{*}$ $p < 0.05$, $^{**}$ $p < 0.01$ significantly different from the AA group; # $p < 0.05$, ## $p < 0.01$ significantly different from the N group.

condition of renal damage or disease. Thus, the plausible mechanisms of methylglyoxal-exacerbating AAN are noted below. First, the binding of AGEs and its receptors upregulates inflammatory effects via nuclear factor kappa-light-chain-enhancer of activated B cells (NF-κB) [47] and early growth response-1 (Egr-1) [48]. Second, methylglyoxal increases oxidative stress, which is assessed via the level of glutathione in the kidneys of mice that received AA due to accumulation of methylglyoxal and $N^{\epsilon}$-(carboxymethyl)lysine (CML) [25]. Third, methylglyoxal-modified collagen might exacerbate interstitial renal fibrosis via the activation of myofibroblasts [49], the inhibition of collagen phagocytosis [50], and changes in the structure of the extracellular matrix [51].

## Conclusion

This study firstly suggested that prednisolone successfully alleviated inflammation and interstitial renal fibrosis and then inhibited glycolysis, which consequently led to reduce methylglyoxal, GLO1, and D-lactate, as well as other glycolysis-related proteins. These findings supported that inhibition of glycolysis might be one of the mechanisms in prednisolone treatment for AAN. These glycolysis parameters (methylglyoxal, GLO1, and D-lactate) increased in the chronic AAN, which were quite different from acute kidney injury model. Since this FD-LC-MS/MS method was indeed helpful for exploring pathological and pharmacological mechanisms, it could be applied to screening disease-related proteins in the future.

## Supporting information

**S1 Table. The gradient elution program of FD-HPLC conditions for protein separation.**
(PDF)

**S2 Table. Altered proteins in the kidney homogenate among three groups on Day14.** [a]The peak numbers correspond to those described in Fig 3. [b]The ratio of AA or AA+P groups to N group showed in the table, and the intensity of N group was regarded as "1". [c]NCBI processed each consecutive sequence record as GI number, a simple series of digits. N, normal group; AA, aristolochic acid group; AA+P: aristolochic acid+prednisolone. *$p < 0.05$, significantly different from AA group; # $p < 0.05$, significantly different from N group.
(PDF)

**S3 Table. Antibody.**
(PDF)

**S1 Fig. The detailed and amplified chromatograms of Fig 5.** The lower, middle, and upper chromatograms were obtained from the kidney homogenates of N- (red), AA- (green), and AA+P- (blue) group mice, respectively. The 47 altered peaks among three groups were numbered.
(PDF)

**S2 Fig. The chromatograms of FD-HPLC for separating proteins.** AA+P indicated the six chromatograms of AA+P group; AA indicated the six chromatograms of AA group; N indicated the six chromatograms of N group.
(PDF)

**S3 Fig. The immunoblotting images for Fig 7.** (A) Both of the partial membranes which blotted GLO1 and β-actin antibodies were acquired from the same gel. After the proteins of gel were transferred onto the nitrocellulose membrane, the membrane was blocked with 10% skim milk. Before incubated with primary antibodies, the membrane was cut into two parts. One partial membrane was incubated with anti-GLO1 antibody, and another was incubated with anti-β-actin antibody, respectively. (B) Both of the band of aldolase B and β-actin were acquired from the same gel and same membrane. (C) Both of the band of TPI and β-actin were acquired from the same gel and same membrane.
(PDF)

**S1 Raw Images. WB beta-actin.**
(TIF)

**S2 Raw Images. WB GLO1.**
(TIF)

**S3 Raw Images. WB Aldolase B & beta-actin.**
(TIF)

**S4 Raw Images. WB TPI & beta-actin.**
(TIF)

## Acknowledgments

We are grateful to the Yung Shin Pharmaceutical Co. (Taiwan) for their contribution of an API 4000 triple quadrupole mass spectrometer. We appreciate that Prof Ueda Shiro supplied our team the suggestion about animal experiments and metabolic cages. We thank for Prof Jen-Ai Lee supplying the technology of FD-LC-MS/MS. We are grateful to the financial support from the Cathay General Hospital (108CGH-TMU-06).

## Author Contributions

**Conceptualization:** Shih-Ming Chen.

**Formal analysis:** Chia-En Lin, Hung-Hsiang Chen, Yu-Fan Cheng.

**Methodology:** Kazuhiro Imai.

**Supervision:** Shih-Ming Chen.

**Writing – original draft:** Shih-Ming Chen, Chia-En Lin, Hung-Hsiang Chen, Yu-Fan Cheng.

**Writing – review & editing:** Shih-Ming Chen, Hui-Wen Cheng, Kazuhiro Imai.

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
