## [Decision Letter · Decision Letter 0]

27 Oct 2019

PONE-D-19-27780

Effect of prednisolone on glyoxalase 1 in an inbred mouse model of aristolochic acid nephropathy using proteomics method with FD-LC-MS/MS

PLOS ONE

Dear Dr. Chen,

Thank you for submitting your manuscript to PLOS ONE. After careful consideration, we feel that it has merit but does not fully meet PLOS ONE’s publication criteria as it currently stands. Therefore, we invite you to submit a revised version of the manuscript that addresses the points raised during the review process.

In particular, try to verify the increase of the Fructose-bisphosphate aldolase B and  TPI by conventional immunological assays and to clarify issues in Figure 5 reported by the reviewer #2. Moreover, again in this figure (panels A and C), the standard deviations of the AA group are very high. Please check wether the differences are statistically significant. All other issues raised by reviewers need to be addressed, as well.

In addition, please note that the present manuscript requires copyediting for English grammar and usage. 

We would appreciate receiving your revised manuscript by Dec 11 2019 11:59PM. To enhance the reproducibility of your results, we recommend that if applicable you deposit your laboratory protocols in protocols.io, where a protocol can be assigned its own identifier (DOI) such that it can be cited independently in the future. For instructions see: http://journals.plos.org/plosone/s/submission-guidelines#loc-laboratory-protocols

We look forward to receiving your revised manuscript.

Kind regards,

Fabio Sallustio

Academic Editor

PLOS ONE

Journal Requirements:

4. We suggest you thoroughly copyedit your manuscript for language usage, spelling, and grammar. If you do not know anyone who can help you do this, you may wish to consider employing a professional scientific editing service. We understand that you have already had your manuscript edited for English, but we still don't consider it to be of a sufficient standard for publication. 

Reviewers' comments:

Reviewer's Responses to Questions

**Comments to the Author**

1. Is the manuscript technically sound, and do the data support the conclusions?

Reviewer #1: Yes

Reviewer #2: Partly

2. Has the statistical analysis been performed appropriately and rigorously? 

Reviewer #1: Yes

Reviewer #2: Yes

3. Have the authors made all data underlying the findings in their manuscript fully available?

Reviewer #1: Yes

Reviewer #2: Yes

4. Is the manuscript presented in an intelligible fashion and written in standard English?

Reviewer #1: Yes

Reviewer #2: Yes

5. Review Comments to the Author

Reviewer #1: The authors studied the effect in terms of altered proteins after prednisolone treatment in a mice model of aristolochic acid nephropathy using a proteomics technique. The authors demonstrated for the first time the biochemical efficacy of prednisolone, and urinary methylglyoxal and its metabolite-D-lactate might be potential biomarkers for aristolochic acid nephropathy. The manuscript shows favorable organization, experiment arrangement, and references. All the data were analyzed and interpreted appropriately. The figures and tables are shown with good quality. Methods described in the manuscript is sufficient to follow. The manuscript contains sufficient data to support its conclusion.

Issues 1: The authors may need to consider polishing the language for a better understanding of international researchers.

Issues 2: The authors need to add more discussion and background in the introduction part.

Minors :

Please use full name in title line.

Please prepare a figure to illustrate the work-flow for the better understanding.

Reviewer #2: COMMENTS TO AUTHORS

Manuscript Number: PONE-D-19-27780

Manuscript Title: Effect of prednisolone on glyoxalase 1 in an inbred mouse model of aristolochic acid nephropathy using proteomics method with FD-LC-MS/MS

Authors: Shih‐Ming Chen, Chia‐En Lin, Hung-Hsiang Chen, Yu-Fan Cheng, Hui-Wen Cheng, Kazuhiro Imai

Comments:

The authors investigate here the effect of prednisolone treatment on the protein expression in a mice model of AAN using a proteomics technique. Six-week‐old female C3H/He mice were randomly divided into three groups: normal (N), aristolochic acid (AA), and aristolochic acid + prednisolone (AA+P) groups (n = 10 each group).The aim of the study is to (1) explore the effect of prednisolone on changes in glycolysis-related protein expression using the FD-LC-MS/MS method and (2) to clarify the pharmacological mechanisms of prednisolone in AA model. The authors found that 1) prednisolone improved tubulointerstitial damage; 2) the glycolysis-related protein glyoxalase 1 (GLO1) decreased in the AA+P group; 3) prednisolone decreased the urinary methylglyoxal which was accompanied with decrease in urinary D-lactate. The authors conclude that 1) The renal protective mechanism of Prednisolone might be associated with down-regulation of GLO1 via reducing the contents of methylglyoxal derived from glycolysis; 2) they demonstrated for the first time the biochemical efficacy of prednisolone, and urinary methylglyoxal and its metabolite-D-lactate might be potential biomarkers for AAN.

This study is well done but some points should be clarified

1) As the authors discussed, the fructose 1,6-bisphosphate is cleaved into glyceraldehyde 3-phosphate (GAP) and dihydroxyacetone phosphate (DHAP) catalyzing by aldolase B. To demonstrate that Prednisolone inhibited glycolysis, which consequently led to reduce methylglyoxal, GLO1, and D-lactate the authors should also verify by a conventional immunological assay the increase of the Fructose-bisphosphate aldolase B and TPI.

2) Its not clear along all the paper how many samples have been used for the different experiments (i.e, how many samples were used for WB analysis?

3) Figure 4

I think , each group chromatogram should be reported with its proper Fluorescence intensity, otherwise chromatograms comparison is not possible.

4) Figure 5

Why did you use peak height instead of peak area to calculate the differences among the 3

different groups of samples? From S1_Fig , peak n. 22 seems higher in Normal (red line) instead of AA (green line); peak n. 28 seems not to be the same in the 3 different chromatograms as well as for peak n.32, which seems to be higher in Normal instead of AA.

5) Figure 8

From Figure 8 A it doesn’t seem that the amount of urinary total lactate in the N group was lower than that in the AA and AA+P groups, instead it seems that the amount of urinary total lactate in the N group was higher than that in the AA and AA+P groups. Please clarify.

6) Table 1

Missing biochemical data at basal (T0) could clarify the differences of the animal groups at the beginning of the study, useful to appreciate and confirm the effect of therap

6. PLOS authors have the option to publish the peer review history of their article (what does this mean?). If published, this will include your full peer review and any attached files.

Reviewer #1: No

Reviewer #2: No

---

## [Author Response · Author response to Decision Letter 0]

5 Dec 2019

Dear reviewers,

Thank you for providing these insights. We have received the letter of 2019/10/27 (PONE-D-19-27780). Our research article was entitled “Effect of prednisolone on glyoxalase 1 in an inbred mouse model of aristolochic acid nephropathy using proteomics method with FD-LC-MS/MS by Shih‐Ming Chen, Chia‐En Lin, Hung-Hsiang Chen, Yu-Fan Cheng, Hui-Wen Cheng, Kazuhiro Imai”. The followings are our response to your suggestion and comment. We highlighted the revised paragraph with red color in “Revised Manuscript with Track Changes”, so that you could figure out the the original and revised paragraph. Please refer in "Response to reviewer" file.

Reviewer #1: 

Issues 1: The authors may need to consider polishing the language for a better understanding of international researchers.

Answer: 

We have corrected the English writing again and provided the certificate from the UNIVERSAL LINK CO., LTD. 

Issues 2: The authors need to add more discussion and background in the introduction part.

Answer:

We have increase the paragraph to introduce the background and discussion in the part of introduction, especially the mechanisms of aristolochic acid-induced cytotoxicity and carcinogenicity from the recent study.

Please use full name in title line.

Answer: 

We have revised our title according to your suggestion, so we have changed our title as the following.

Please prepare a figure to illustrate the work-flow for the better understanding.

Answer: 

We have prepared a flow chart to explain for our study step by step in Fig 1

Reviewer #2: COMMENTS TO AUTHORS

(1)As the authors discussed, the fructose 1,6-bisphosphate is cleaved into glyceraldehyde 3-phosphate (GAP) and dihydroxyacetone phosphate (DHAP) catalyzing by aldolase B. To demonstrate that Prednisolone inhibited glycolysis, which consequently led to reduce methylglyoxal, GLO1, and D-lactate the authors should also verify by a conventional immunological assay the increase of the Fructose-bisphosphate aldolase B and TPI.

Answer: 

Thanks for your suggestion. We have performed the immunoblotting for aldolase B and TPI to confirm protein expression. Moreover, we have acquired the similar results to the proteomics findings. We have added these new findings in the part of “Method and materials” and “Results”.

(2) Its not clear along all the paper how many samples have been used for the different experiments (i.e, how many samples were used for WB analysis?

Answer: 

All mice of biochemical data (including BUN, Scr, urinary protein, and NAG), sections (PAS stain, Masson trichrome stain, and immunofluourance), urinary methylglyoxal, and urinary D-lactate were determined. We selected six mice from each group to perform proteomics study. We selected six mice from each group to perform immunoblotting for glyoxalase1, aldolase B, and triphosphate isomerase because of the rest of the kidney tissue.

(3) Figure 4

I think , each group chromatogram should be reported with its proper Fluorescence intensity, otherwise chromatograms comparison is not possible.

Answer:

We have reported the chromatograms according to your suggestion. The chromatograms with the correct scales were shown in Fig 5.

4) Figure 5

Why did you use peak height instead of peak area to calculate the differences among the 3 different groups of samples? From S1_Fig , peak n. 22 seems higher in Normal (red line) instead of AA (green line); peak n. 28 seems not to be the same in the 3 different chromatograms as well as for peak n.32, which seems to be higher in Normal instead of AA.

Answer:

Usually, peak area is better than peak height. In our case, each peak has different baseline. If we calculated with peak area, the intensity of the peak might include part of other peak area. This FD-LC-MS/MS method used peak height rather than peak area according to the previous study (Ichibangase et al., Journal of Proteome Research; 2007, 6, 2841-2849).

About this question “From S1_Fig , peak n. 22 seems higher in Normal (red line) instead of AA (green line); peak n. 28 seems not to be the same in the 3 different chromatograms as well as for peak n.32, which seems to be higher in Normal instead of AA.”, we have labeled peak no.22, 28, and 32 of each chromatogram of sample in each group in S2_Fig. According to the results of S2_Fig, we acquired the intensity and expressed as mean ± SD in Table 2 and Table S2. All of this intensity is not only acquired from S1_Fig. Thus, we average the intensity of six sample data from each group and compared among three groups.

5) Figure 8

From Figure 8 A it doesn’t seem that the amount of urinary total lactate in the N group was lower than that in the AA and AA+P groups, instead it seems that the amount of urinary total lactate in the N group was higher than that in the AA and AA+P groups. Please clarify.

Answer:

Answer: Figure 8 A and 8C indicated the “level” of total lactate according to the chromatograms but not the “amount” of total lactate. Figure 8B and 8D indicated the “level” of D-lactate according to the chromatograms but not the “amount” of D-lactate. The amount of total lactate was expressed as: amount of lactate (mmol) = level of lactate (mM) x urinary volume (µL); Expression of D-lactate was similar to expression of total lactate (amount of D-lactate [µmol] = level of lactate [µM] x urinary volume [µL]). Because the level of lactate is very low, we use the amount which could more reflect how much methyglyoxal metabolize into D-lactate in mouse body.

6) Table 1

Missing biochemical data at basal (T0) could clarify the differences of the animal groups at the beginning of the study, useful to appreciate and confirm the effect of therap

Answer: We have determined the urinary data (urinary protein and NAG) on baseline according to your suggestion. However, it is not impossible to acquire the blood data on baseline, because it is necessary to acquire more than 60 µL of blood sample so that we could analyze the data. The body weight of mouse is about 20 g, and it is a high stress to get such volume of blood for a mice. We only can get such large volume, when the mouse were sacrificed. In order to solve this problem, we compared blood data of the “normal group” with AA and AA+P groups on day 14.

---

## [Decision Letter · Decision Letter 1]

31 Dec 2019

Effect of prednisolone on glyoxalase 1 in an inbred mouse model of aristolochic acid nephropathy using a proteomics method with fluorogenic derivatization-liquid chromatography-tandem mass spectrometry

PONE-D-19-27780R1

Dear Dr. Chen,

We are pleased to inform you that your manuscript has been judged scientifically suitable for publication and will be formally accepted for publication once it complies with all outstanding technical requirements.

With kind regards,

Fabio Sallustio

Academic Editor

PLOS ONE

Additional Editor Comments (optional):

Reviewers' comments:

Reviewer's Responses to Questions

**Comments to the Author**

1. If the authors have adequately addressed your comments raised in a previous round of review and you feel that this manuscript is now acceptable for publication, you may indicate that here to bypass the “Comments to the Author” section, enter your conflict of interest statement in the “Confidential to Editor” section, and submit your "Accept" recommendation.

Reviewer #1: All comments have been addressed

2. Is the manuscript technically sound, and do the data support the conclusions?

Reviewer #1: Yes

3. Has the statistical analysis been performed appropriately and rigorously? 

Reviewer #1: Yes

4. Have the authors made all data underlying the findings in their manuscript fully available?

Reviewer #1: Yes

5. Is the manuscript presented in an intelligible fashion and written in standard English?

Reviewer #1: Yes

6. Review Comments to the Author

Reviewer #1: (No Response)

7. PLOS authors have the option to publish the peer review history of their article (what does this mean?). If published, this will include your full peer review and any attached files.

Reviewer #1: No

---

## [Editor Report · Acceptance letter]

7 Jan 2020

PONE-D-19-27780R1 

Effect of prednisolone on glyoxalase 1 in an inbred mouse model of aristolochic acid nephropathy using a proteomics method with fluorogenic derivatization-liquid chromatography-tandem mass spectrometry 

Dear Dr. Chen:

I am pleased to inform you that your manuscript has been deemed suitable for publication in PLOS ONE. Congratulations! Your manuscript is now with our production department. 

With kind regards,

on behalf of

Dr. Fabio Sallustio 

Academic Editor

PLOS ONE